# The lived experience of long COVID: A qualitative study of mental health, quality of life, and coping

Colleen E. Kennelly[1,2], Anh T. P. Nguyen[2], Natasha Yasmin Sheikhan[1,2], Gillian Strudwick[1,2], Chantal F. Ski[3], David R. Thompson[3], Mary Bartram[4], Sophie Soklaridis[1,2], Susan L. Rossell[5], David Castle[6], Lisa D. Hawke[1,2]*

1 University of Toronto, Toronto, Ontario, Canada, 2 Centre for Addiction and Mental Health, Toronto, Ontario, Canada, 3 Queen's University Belfast, Belfast, Northern Ireland, United Kingdom, 4 Mental Health Commission of Canada, Ottawa, Ontario, Canada, 5 Swinburne University of Technology, Melbourne, Victoria, Australia, 6 University of Tasmania, Hobart, Tasmania, Australia

* Lisa.Hawke@camh.ca

**Data Availability Statement:** The data underlying the results presented in the study are available from the Centre for Addiction and Mental Health upon request, review, and approval by the

## Abstract

The majority of people who contract COVID-19 experience a short period of symptomatic infection. However, symptoms persist for months or years following initial exposure to the virus in some cases. This has been described as long COVID. Little is known about the lived experience of this condition, as it has only recently emerged. This study aimed to explore the experiences of mental health, quality of life, and coping among people living with long COVID. A sample of 47 adults with lived experience participated in web-based focus groups. Separate focus groups were held for 24 individuals with pre-existing mental health conditions and 23 individuals without pre-existing mental health conditions. Data were analyzed using a codebook thematic analysis approach. Five themes were identified as integral to the long COVID experience: The Emotional Landscape of Long COVID, New Limits to Daily Functioning, Grief and Loss of Former Identity, Long COVID-related Stigmatization, and Learning to Cope with Persisting Symptoms. These findings illustrate the immense impact of long COVID on mental health and quality of life. Minimal differences were found between those with and those without pre-existing mental health conditions, as both groups were substantially impacted by the condition. Attention to the perspectives of people with lived experience of long COVID is necessary to inform future directions for research and clinical practice.

## Introduction

The majority of people who contract SARS-CoV-2 (80%) are asymptomatic or experience mild symptoms for a short period of time, while a small proportion report severe symptoms (15%) or become critically ill (5%) during acute infection [1]. Despite this variation, all people who have been exposed to SARS-CoV-2 are at risk of developing an enduring set of symptoms–known as long COVID–which may linger for months or years [2]. The World Health

Research Ethics Board (Research.Ethics@camh. ca).

**Funding:** This study was supported by funding awarded to LDH from the Canadian Institutes of Health Research (Funding #: WI1-179893). The funder was not involved in study design, data collection and analysis, preparation of the manuscript, or the decision to publish.

**Competing interests:** David Castle has received grant monies for research from Servier, Boehringer Ingelheim; Travel Support and Honoraria for Talks and Consultancy from Servier, Seqirus, Lundbeck. He is a founder of the Optimal Health Program (OHP), and holds 50% of the IP for OHP; and is part owner of Clarity Healthcare. He does not knowingly have stocks or shares in any pharmaceutical company. Other authors have no conflict of interest to declare. This does not alter our adherence to PLOS ONE policies on sharing data and materials.

Organization (WHO) defines long COVID as a condition in which symptoms emerge or persist three months after a confirmed or suspected case of SARS-CoV-2 infection and endure for upwards of two months in the absence of an alternative explanation [3]. Recent evidence emerging from a systematic review and meta-analysis indicates approximately 45% of people infected with SARS-CoV-2 will experience at least one unresolved symptom four months after infection, regardless of hospitalization status [4]. Risk factors for long COVID include being female, belonging to a racialized group, socioeconomic deprivation, obesity, smoking, and the presence of one or more conditions affecting physical or mental health [5, 6].

Although SARS-CoV-2 is a respiratory virus, long COVID has a multisystem impact [7]. Over 50 symptoms have been described in association with long COVID, most commonly: fatigue, shortness of breath, cough, chest pain, muscle and joint pain, fever, the loss of the sense of smell and taste, hair loss, and sexual dysfunction [6, 8, 9]. Symptoms may change over time, as is exemplified by reports of cough and loss of taste becoming less prevalent while muscle pain becomes more prevalent [10].

In addition to physical symptoms, preliminary research indicates the substantial impact of long COVID on mental health and quality of life. These effects compound the decline in mental health observed among the general public during the COVID-19 pandemic [11]. Specifically, long COVID is associated with the onset or worsening of neurological or mental health conditions including attention-deficit/hyperactivity disorder, post-traumatic stress disorder (PTSD), depression, anxiety, memory loss, brain fog, and insomnia [6, 8, 12]. In a longitudinal cohort study examining health outcomes two years after acute infection with SARS-CoV-2, individuals with long COVID symptoms reported worse mental health outcomes and health-related quality of life compared to both COVID-19 survivors without long COVID symptoms and the general public [13]. Moreover, findings from a naturalistic qualitative study indicate that some individuals experiencing long COVID have difficulty returning to pre-infection routines and activities given the persistence of physical and mental health symptoms that impede daily functioning [14].

Lived experience perspectives in long COVID research are important for establishing knowledge that accurately represents the human impact of this complex condition. Recently, a small but growing collection of qualitative research examining the long COVID experience has emerged. These include inquiries into the experience of long COVID and access to health-care services [14–16], the experience of doctors as long COVID patients [17], and an exploration of the role of physical activity in the long COVID experience [18]. Recent systematic reviews of the long COVID qualitative literature highlight the breadth of challenges faced by those with lived experience, including the persistence of symptoms, emotional distress, shifting social supports, and barriers to health care services [19, 20]. Most of these studies to date originated from the United Kingdom, with a high representation of participants who were young or middle-aged adults, female-identified, white, and recruited online [19, 20].

Emerging from the Canadian context, this study builds upon the pre-existing literature by engaging a diverse group of adults living with long COVID in focus group discussions. Specifically, the purpose of this study is to gain more specific insight into the mental health, quality of life, and coping experiences of people living with long COVID through qualitative exploration. The impact of a past history of mental health conditions was also examined.

## Methods

This is a descriptive qualitative study embedded in a larger patient-oriented research study which consists of investigations into: (1) the mental health, quality of life, and coping experiences of people living with long COVID; (2) long COVID treatment experiences and

preferences; and (3) the experiences of long COVID service providers. This study was approached through a pragmatic lens, given the compatibility of patient-oriented research on the experience of long COVID with associated paradigmatic principles of democratic values, problem-solving through collaboration, and social justice [21]. Study procedures were conducted in compliance with the Canadian Institutes of Health Research Strategy for Patient-Oriented Research [22]. The Consolidated Criteria for Reporting Qualitative Research (COREQ) provides the framework for reporting the methods and results [23], while engagement is reported using the Guidance for Reporting Involvement of Patients and the Public– 2 (GRIPP2) short checklist [24].

## Lived experience advisory group

Proactive engagement of lived experience was achieved in this study through the assembly of a 'lived experience advisory group' comprised of six individuals experiencing long COVID. This group actively advised on study protocol and outcomes from the initial stages through to study completion. The aim, methods, and results of lived experience engagement are summarized in Table 1.

## Participants

Eligible participants were aged 18 or older, proficient in English, and self-identified as experiencing long COVID in accordance with the WHO definition [3]. We attempted to fill pre-specified quotas in the sampling process, with some flexibility to account for

**Table 1. GRIPP2 reporting checklist for the engagement of people with lived experience in research.**

| Section & topic | Description |
|---|---|
| **1: Aim** | People living with long COVID were engaged to ensure research questions, interpretations of data, and reporting of findings accurately represented the lived experience with long COVID. |
| **2: Methods** | A lived experience advisory group comprised of six people was initially assembled, although one advisor formally withdrew from the group and one advisor failed to respond to correspondence before study completion. At the time of completion of this manuscript, the lived experience advisory group had assembled for seven meetings and influenced decision making associated with numerous aspects of study development and execution. |
| **3: Study results** | The lived experience advisory group accomplished many tasks throughout the engagement period. These included providing feedback on the semi-structured interview guide, advising on recruitment materials, utilizing social networks to bolster recruitment, ensuring study findings accurately represented the long COVID experience, and assisting with the development of knowledge translation materials. |
| **4: Discussion and conclusions** | The lived experience advisory group made considerable contributions to the study. Notably, advisors improved the interview guide and bolstered recruitment efforts by promoting the study among their connections within the long COVID community. Further, advisors provided insight into the interpretation and relevancy of early codes, theme development, and quote selection, from a lived experience perspective. This feedback was fundamental to informing to the process of drafting the manuscript. |
| **5: Reflections and critical perspective** | The success of this study would not have been possible without the lived experience advisory group. However, challenges associated with lived experience engagement included attrition and the inability to incorporate all advisor feedback due to conflicting views or other limitations, which were openly discussed with the group. Consideration of attrition when recruiting lived experience advisors and the fostering of open and honest communication between the research team and the advisory group minimized the impact of these challenges, resulting in an overwhelmingly positive experience with lived experience engagement. |

intersectionality. We aimed to achieve diversity in terms of racialized groups; representation of both women and men; representation of people with transgender or gender expansive identities; diversity across age including younger adults (<35 years), mid-age adults (35–59 years) and older adults (>60 years); and Canada-wide recruitment. Purposive sampling was also employed to ensure approximately equal representation of participants with and without pre-existing mental health conditions. Of the 67 potential participants screened, seven were excluded as one was not actively experiencing long COVID symptoms, two had not contracted COVID-19 at least three months prior to screening, three were outside of demographic quotas, and one withdrew during the screening process. A further eight prospective participants completed screening and five provided consent, but ceased to respond to study correspondence or were unable to attend a scheduled focus group, resulting in a final sample size of 47. Participants did not have pre-existing relationships with any member of the research team.

## Procedures

Recruitment occurred between June and December 2022. Flyers were circulated via email within the research team's networks, among institutional research partners, and to long COVID clinics and community support organizations across Canada. Recruitment information was published on internal and external institutional websites and advertised on social media at the Centre for Addiction and Mental Health (CAMH). Interested individuals connected with the research team over phone, email, or text. After screening for eligibility, prospective participants attended a virtual session to learn more about the study and consented to participate. Consented participants completed a demographic survey collected using REDCap electronic data capture tools hosted at CAMH [25, 26]. Six focus groups included 24 participants who had not experienced pre-existing mental health conditions, while five were comprised of 23 participants who had experienced pre-existing mental health conditions. Participants received a $50 gift card for full participation or a $25 gift card if they attended half the focus group.

## Ethical considerations

All participants provided written informed consent using digital signatures. Approval for this study was granted by the Research Ethics Board of the Centre for Addiction and Mental Health (030–2022).

## Data collection

Data from 11 focus groups conducted over a secure Webex videoconferencing system were collected between August and December 2022. Each 60–90 minute focus group was co-facilitated by a lead facilitator with a bachelor's education (ATPN) and either a post-doctoral research fellow or doctoral student (NYS). Support from clinical personnel was available, but not requested. Facilitators moderated the discussions using a semi-structured interview guide that was developed in consultation with the lived experience advisory group and pilot tested by members of a research unit at CAMH specializing in lived experience engagement. The interview guide consisted of 20 questions, covering: (1) mental health, quality of life, and coping; (2) treatment experiences and preferences; and (3) equity, diversity, and inclusion (EDI) factors. The interview guide differed slightly for those with or without pre-existing mental health conditions, to inquire into how coping and experiences with long COVID were influenced by pre-existing mental health conditions. Participants contributed to the sessions using Webex microphone or chat functions. Facilitators read written statements aloud to ensure recognition by the group. Each session was recorded and transcribed verbatim by a professional

transcriptionist or member of the research team and proofed by a separate research team member. Transcripts were uploaded to NVivo (Ver 12) for analysis [27]. Authors responsible for recruitment, data collection, or analysis (CEK, ATPN, NYS, LDH) had access to information that could identify individual participants during and after data collection.

## Data analyses

A codebook thematic analysis method modelled after the combined inductive and deductive approach of Fereday and Muir-Cochrane [28] was employed. A codebook was developed collaboratively by the lead focus group facilitator (APTN) and team members based on prior knowledge of literature relating to the research questions and initial scans of the data, before being refined through discussion with the analysis team. The primary analyst (CEK) became familiar with the transcripts through multiple readings, which informed iterative changes to the codebook. The transcripts were then systematically reviewed in detail, and the coding framework was applied to the text. New codes were identified inductively and discussed with ATPN and LDH before they were added to the codebook. Themes were developed upon review of both codes and the original text. Tentative codes and themes were brought to the lived experience advisory group to ensure accurate representation. Steps taken to build trustworthiness included the practice of reflexivity through internal dialogue and discussions among the analysis team, extensive engagement with transcripts, ongoing consultation with research team members and the lived experience advisory group, documentation of team meetings, maintenance of audit trails, and transparency regarding methodological decision-making [29].

## Research team positionality

The first author and analyst is a white female-identified Research Trainee living in Canada. She is a student pursuing a Master of Public Health in Health Promotion and has not experienced long COVID. The second author and lead focus group facilitator is a female-identified, South East Asian immigrant in Canada. She is a Research Analyst with a Bachelor of Arts in Psychology. While not experiencing long COVID, she has an uncle who has long-term cognitive and emotional symptoms after severe acute COVID-19 infection. They are supported by an interdisciplinary team of researchers at diverse career stages and with diverse perspectives.

## Results

Participant demographics are presented in Table 2. Participants were 24 to 69 years old, with an average age of 44.9 years (SD = 13.0). The majority identified as women (59.6%), white (57.4%), residents of Ontario (53.2%), and employed (59.6%). Only four (8.5%) participants identified with each of these characteristics in combination, demonstrating intersectionality. Twenty-three (48.9%) reported pre-existing mental health conditions. Although all participants reported experiencing long COVID symptoms at enrollment, only 23 (48.9%) had received a formal clinical diagnosis of long COVID.

Participants' experiences related to mental health, quality of life, and coping are shown through five themes: *The Emotional Landscape of Long COVID*, *New Limits to Daily Functioning*, *Grief and Loss of Former Identity*, *Long COVID-related Stigmatization*, and *Learning to Cope with Persisting Symptoms*. Quotes contributed by participants with pre-existing mental health conditions are labelled with "Pre-MH", while those belonging to participants without pre-existing mental health conditions are labelled with "No pre-MH". Themes and subthemes are summarized in Table 3.

**Table 2. Demographic characteristics of study participants.**

| Demographic characteristics | Participants (n = 47) | Percent (%) |
|---|---|---|
| **Age** | | |
| 18–34 | 9 | 19.1 |
| 35–55 | 20 | 42.6 |
| >55 | 11 | 23.4 |
| Missing | 7 | 14.9 |
| **Gender identity** | | |
| Man (cis or transgender) | 18 | 38.3 |
| Woman (cis or transgender) | 28 | 59.6 |
| Non-binary | 1 | 2.1 |
| **Ethnicity** | | |
| White | 27 | 57.4 |
| Indigenous | 6 | 12.8 |
| South Asian | 4 | 8.5 |
| East or Southeast Asian | 2 | 4.3 |
| More than one ethnicity | 3 | 6.4 |
| Another ethnicity [a] | 3 | 6.4 |
| Missing | 2 | 4.3 |
| **Place of residence** | | |
| Ontario | 25 | 53.2 |
| Western Canada | 14 | 29.8 |
| Quebec | 4 | 8.5 |
| Eastern Canada | 3 | 6.4 |
| Missing | 1 | 2.1 |
| **Employment status** | | |
| Employed | 28 | 59.6 |
| On disability/sick leave | 9 | 19.1 |
| Unemployed | 4 | 8.5 |
| Retired | 4 | 8.5 |
| Other | 2 | 4.3 |
| **Time since COVID-19 infection** | | |
| 3–6 months | 9 | 19.1 |
| 7–12 months | 15 | 31.9 |
| 13+ months | 17 | 36.2 |
| Missing | 6 | 12.8 |
| **Self-reported physical health** | | |
| Good to excellent | 22 | 46.8 |
| Fair to poor | 25 | 53.2 |
| **Self-reported mental health** | | |
| Good to excellent | 14 | 29.8 |
| Fair to poor | 33 | 70.2 |
| **Pre-existing Mental Health Condition** | | |
| Yes | 23 | 48.9 |
| No | 24 | 51.1 |
| **Receipt of a long COVID diagnosis** | | |
| Yes | 23 | 48.9 |
| No | 23 | 48.9 |

(*Continued*)

**Table 2.** (Continued)

| Demographic characteristics | Participants (n = 47) | Percent (%) |
|---|---|---|
| Unsure | 1 | 2.1 |

<sup>a</sup> Includes ethnicities that are not otherwise specified

[a] Includes ethnicities that are not otherwise specified

## Theme 1: The emotional landscape of long COVID

Participants reported declining emotional health and wellbeing following the onset of long COVID. This shift in emotional state was heavily tied to the lingering and unpredictable nature of the physical and mental health symptoms that came with long COVID. These effects were reported by those with and without pre-existing mental health conditions alike, with many participants from both groups acknowledging their mental health had been well managed prior to long COVID. Subthemes represent this experience: anxiety and fear of the unknown, challenges with emotional regulation, and feelings of hopelessness and depression.

**Anxiety and fear of the unknown.** Participants discussed feeling a heightened sense of anxiety as a symptom of long COVID. This experience was shared amongst those with and without pre-existing anxiety disorders. For instance, one participant reflected on how long COVID impacted their pre-existing anxiety, stating "I already had severe anxiety before I got COVID but the symptoms with long COVID had definitely made it worse." (Pre-MH) Other participants felt that long COVID caused anxiety, as is described by the following participant:

"The anxiety that came because of [long COVID]—I don't know what it does or how it triggers your brain, if it does it at all, but I believe that it did something that created anxiety." (No pre-MH)

Many participants discussed feeling anxious about the unpredictable nature of long COVID and were fearful of how new or worsening symptoms might impact their health, finances, and social life. One participant used a 'rollercoaster ride' as a metaphor for their fear about the unpredictable nature of long COVID, recounting how they lived in constant terror and uncertainty around what each day would look like for them. Others described a fear of setbacks in their recovery, for instance:

"I've been very isolated for the past year, as it's only really been the past six months that I've started getting out again. And that's been really challenging, and I've not had anxiety from

**Table 3. Summary of themes and subthemes.**

| Themes | Subthemes |
|---|---|
| The Emotional Landscape of Long COVID | Anxiety and fear of the unknown |
| | Challenges with emotional regulation |
| | Feelings of hopelessness and depression |
| New Limits to Daily Functioning | Living with brain fog |
| | Exhaustion as a barrier to completing tasks in everyday life |
| Grief and Loss of Former Identity | Reconciling past and present identities |
| | Loss of social connections |
| Long COVID-related Stigmatization | Trivialization of experience with long COVID |
| | The burden of blame |
| Learning to Cope with Persisting Symptoms | Establishment of new habits affecting the health of mind and body |
| | Social support and long COVID knowledge sharing |

social settings before. . .I'm just so worried about getting COVID again and all that progress evaporating and having to start over again." (Pre-MH)

**Challenges with emotional regulation.** Participants reported difficulty with emotional regulation as a symptom of long COVID, causing them to experience sudden and uncontrollable episodes of sadness or rage. One participant recounted their experience with rapidly shifting mood, sharing "For me, it's the emotional sensitivity. I get angry at the drop of a hat, sad at the drop of a hat. I mean, there's a million commercials I can't watch." (Pre-MH) While some participants believed these difficulties with emotional regulation to be a direct effect of long COVID, others connected their short temper to their frustrations with long COVID symptoms. To illustrate, one participant reflected on how their low energy levels have contributed to feelings of irritability:

"My energy levels are, as I mentioned, very low. So. . .my threshold for any stress, it's very short, very irritable, because I'm not able to do stuff. So I'm irritable. If anybody [asks] me a simple question, I'm upset at them." (No pre-MH)

**Feelings of hopelessness and depression.** Participants reported experiencing feelings of depression and a sense of hopelessness about the persistence of long COVID symptoms for months or years after initial exposure to the virus. This was discussed among participants with and without previous experiences with depression, and was often connected to a lack of hope that long COVID symptoms would ever resolve. For example, one participant reported a deepening sense of hopelessness after three years of long COVID:

"I'm a lifelong survivor of bipolar disorder. I have attempted suicide several times. I'm on medication to control my dark spots that I get. . .it's quite a challenge to live with this. It catapulted me into another dimension. I want to come back, but I can't. It's been three years and I don't see anything concrete, any results that prove that it's going to change." (Pre-MH)

For some participants, these feelings of hopelessness and depression contributed to thoughts of death. For instance, one participant reported engaging in suicidal ideation due to the life-altering impact of long COVID, stating "It has completely ruined my life and I'm 10 seconds away from suicide at any time. I don't have any money anymore, I don't have friends anymore, and I am always in pain." (Pre-MH) Another participant discussed their contemplation of medical assistance in dying (MAID) in the absence of progress towards recovery from long COVID, sharing:

"This long COVID, for myself, if I can't feel that I am coming out of it. . .if things get worse and worse and worse, I would opt for medical assistance in dying. I don't want to be—it's hard to go through this life every day. Takes so much energy that I don't have to just breathe." (No pre-MH)

## Theme 2: New limits to daily functioning

Participants reported new limits to daily functioning due to long COVID, as physical and cognitive symptoms made it difficult for some to complete daily activities and routines practiced prior to infection. These limitations are discussed across two subthemes: living with brain fog and exhaustion as a barrier to completing tasks in everyday life.

**Living with brain fog.** Participants expressed frustration with brain fog as a symptom of long COVID, as effects such as poor memory, confusion, and inability to focus impacted their ability to complete daily tasks. Some described the cognitive impact of long COVID as debilitating, and reported it was a struggle for their brain to complete basic tasks in daily life. Others felt the lingering effects of brain fog predominantly at work, with one participant explaining:

"For me, I am finding it's hard to concentrate and get down to my work. And memory issues as well. I have to write everything down and sometimes even then I don't even get it right. I am finding that it takes me longer to do things than it would normally." (No pre-MH)

Participants also reported brain fog resulting from long COVID as a hazard to themselves and others. For instance, some participants became forgetful or confused when engaging in potentially hazardous behaviours, such as cooking, resulting in the self-restriction of tasks which could not be completed safely on an independent basis. This type of confusion in a high-risk setting was also reported while driving, with one participant stating:

"I could understand there is a vehicle in front of me or if the light is green or if the light is red. I can see that it is red, but I may not understand what that means so I don't know where I am supposed to go or what I am supposed to do or 'do I drive' or 'do I not'." (Pre-MH)

**Exhaustion as a barrier to completing tasks in everyday life.** Participants reported extreme physical exhaustion, which reduced their ability to complete activities in their everyday life. Many participants found they needed to complete tasks in short bursts balanced with periods of rest to avoid post-exertional malaise, as is exemplified by the following reflection:

"Physically, I can't do a whole lot at once. It's a little bit throughout the day. I've finally started making some meals, nothing extravagant. I can't do housekeeping the way I [used] to, or I have to bunch it up and alternate days. Otherwise I'm toasted." (Pre-MH)

A reduced threshold for in-person and remote forms of socialization was frequently acknowledged by participants, often attributed to the combined effects of the emotional and physical exhaustion that came with long COVID. One participant noted a departure from their former social habits due to the immense exhaustion associated with long COVID, sharing "I used to be super social, and have people over all the time. . .I just don't have the energy mentally, emotionally to do people." (No pre-MH) This sentiment was also true for those engaging in virtual socialization, with one participant sharing:

"Emailing with friends—just social emailing. It takes me weeks and sometimes even months to get back to somebody because I just don't want to deal with it. I am so exhausted. Even composing an email is exhausting to me. It's the mental exhaustion in addition to the physical exhaustion." (No pre-MH)

## Theme 3: Grief and loss of former identity

Participants reported grieving the person they were before long COVID. They also experienced loss of elements of their former identity, including work, relationships, and hobbies. Each of these fragments of identity are considered to be protective factors for mental health, changes to which led to a decline in wellbeing and quality of life for many participants.

Subthemes represent the breadth of these feelings of loss: reconciling past and present identities and loss of social connections.

**Reconciling past and present identities.** Participants reported grieving the loss of their former self and had difficulty relating to the person they had become through the long COVID experience. For instance, one participant explained the emotional impact of reconciling past and present identities under long COVID:

> "Some days I had very, very, very dark thoughts because I just absolutely did not see how it was going to truly resolve, like any of the situations. . .it's hard to see a real positive outcome when you're struggling to relate to your own self and to rethink your entire life, in my case." (Pre-MH)

Shifting perceptions of identity were also directly connected to experiences of reduced productivity due to long COVID symptoms, particularly at work. This was a difficult change to reconcile for many participants, as productivity was central to their former sense of self. One participant described the sudden and devastating loss of productivity using the metaphor of 'falling off a cliff':

> "I'm someone who's very productive. I retired in 2017, but since then I haven't stopped. I've been involved in volunteer activities. I was on three boards for different artistic organizations across the city. I was working on my own project that I presented in May. I got COVID in June. . .and it was like falling off a cliff. I wasn't able to be productive and to contribute to all the things I'm involved in the way I did before." (Pre-MH)

Others discussed struggling to regain productivity at work after sick leave, but ultimately had to let go given limitations imparted by long COVID symptoms. For example, one participant recounted their process of reconciling elements of their identity tied to work:

> "I tried to go back to work for a while. I thought maybe I can just fight through this and it'll be alright. Just push through and push through, but it just didn't go away. It just didn't get better. And then I just kept trying to sort of push through, fight through, and I just said forget it. It's just not worth it. It's not worth it to work and stuff just because I can't concentrate, really. I can do menial tasks kind of work, but that's not what I'm sort of known for." (No pre-MH)

The ability of participants to engage in joyful pursuits was also impacted by long COVID symptoms, and this further contributed to feelings of reduced productivity and shattered sense of self. This is exemplified by the experience of a participant who was able to return to work, but found they had little energy left for aspects of life that once brought them joy:

> "I was off work for a couple of months. Then when you go back to work, work is all you can do. And then you just didn't do what you enjoy or any of the social piece that you used to do because it took everything to kinda get through work." (No pre-MH)

**Loss of social connections.** Many participants noted loss of connection to others while coping with long COVID as a substantial change to their identity. Some individuals acknowledged their inability to work while coping with long COVID contributed to difficulties with socialization, for instance:

"I just found I couldn't cope at work, I couldn't do my work, I didn't want to. And I have been doing this for years. That then with the isolation from work. . .Everything was work [associated] so when my work was gone, everything else was sort of gone. My connection to people, but I didn't wanna be around people." (No pre-MH)

Others became socially withdrawn due to a combination of symptoms and the perceived inability of social connections to understand their situation. For example, a participant recounted their experience of becoming socially withdrawn from friends due to their lack of understanding of the long COVID experience:

"Plus losing quite a few friends over this entire situation. Not because there's anything wrong with them. It's just that I withdrew, and they couldn't really understand. And I think people just eventually kind of give up on it asking whether or not you're coming out or whether or not you're interested in doing something." (Pre-MH)

Others became socially disconnected from their family due to the burden of long COVID symptoms, which prevented them from engaging in daily activities or major milestones. One participant described this experience as if they were a passive observer to their own life, unable to keep pace with their family:

"It has really impacted how I deal with family and home. . .I certainly have mood swings more than I have ever had in my entire life. And I don't know if that is a post-COVID issue, is there a combination of things, not being able to have normalcy in my life. And how I am feeling and always been in a constant distress. . .I have got little ones at home and it's tough. Mom looks after them for the course of the day and I am kind of like that vegetable [in] the corner and it feels like life is passing me by in that sense." (Pre-MH)

### Theme 4: Long COVID-related stigmatization

Participants reported diverse experiences with stigma from others and self-stigma while coping with long COVID. Stigma was described as manifesting in negative or unfair beliefs or attitudes towards the existence of long COVID and the pace of recovery from the condition. Two subthemes represent these experiences: trivialization of experience with long COVID and the burden of blame.

**Trivialization of experience with long COVID.**   Participants reported feeling stigma from people who trivialized or invalidated their experience with long COVID due to the belief that that long COVID does not exist or is not a severe illness. This is exemplified by the following contribution, which details a participant experience with trivialization from family and co-workers:

"I do notice it quite a bit through family and workplace. Really anybody you come in contact with who hasn't experienced it the way you have. . .Most people think oh, it's just the common cold and flu and it's all in your head. You know, so when I hear people say it's not real or it's just a flu, I am hearing that they tell me I am not real." (No Pre-MH)

Some participants with pre-existing mental health conditions felt their experience with long COVID was trivialized due to their medical history. One participant described frustrations with the compounding effects of mental health stigma and long COVID stigma, stating:

"I would say the only stigma I have experienced is that I have already had a mental illness before that and that they don't take me seriously enough because it's just brushed off as like

something that I am whining about. Like—it's just—I often wonder if it would be taken more seriously if I haven't had mental illness going into this experience." (Pre-MH)

Stigmatization was also felt in moments when participants were told they should be recovering from long COVID more quickly. This source of long COVID-related stigma was verbalized by a participant who shared:

"One is from friends, in a way it is kind of a stigma. . . like, 'Oh, you should be better by now. Well, you look better today. You don't look so bad. You'll get over it.' And it's like, yeah, I want to. I'm doing everything I think I ought to be doing to do that, but it's taking time." (Pre-MH)

This trivialization of the impact of persistent long COVID symptoms led to self-stigma, expressed through minimization of personal experience with the condition. This process was illustrated by a participant, noting "So many people think it's nothing and so there is a struggle with that. That makes you feel like you have to minimize it a bit." (No pre-MH)

**The burden of blame.**    Participants felt stigmatized by those who blamed them for their own illness, leading to practices of self-censorship and concealment of the lived experience of long COVID to avoid negative perception and treatment from others. One participant discussed their frustrations with these assumptions of blame, stating:

"I think that people kind of wanna say 'well, you got COVID, so you have to deal with the repercussions of it,' as if I didn't do everything to avoid in the first place. And even if I hadn't done everything, there are people who are out there experiencing these symptoms that weren't able to do those things [to protect] themselves. It's still not their fault they are experiencing the long term COVID symptoms." (No pre-MH)

Some participants described feelings of anger and internalized blame in the absence of an alternative explanation for their experience with long COVID, with one participant sharing: "I'm angry and that's not gonna change for a very long time because there's no one to blame, which means it's my fault." (Pre-MH) Other participants questioned their own state of mind amidst the uncertainty of symptoms and long COVID itself, for instance:

"For me, the challenges are largely invisible. And I spend a significant amount of my time questioning whether I'm lost my mind completely or whether what I'm experiencing can be attributed to what's now known as long COVID. And I don't like second guessing myself." (Pre-MH)

### Theme 5: Learning to cope with persisting symptoms

Participants reported a variety of informal mechanisms used to cope with long COVID in the absence of formalized treatment or guidance from healthcare providers. Though the process of researching and testing methods of coping was frustrating, many participants found some success in identifying practices that improved their experience with long COVID. Methods of coping are summarized in the following subthemes: establishment of new habits affecting the health of mind and body and social support and long COVID knowledge sharing.

**Establishment of new habits affecting the health of mind and body.**    Participants described mindset changes as essential to coping with long COVID. While most emphasized the importance of acceptance and optimism, others acknowledged the value of gratitude and

faith in their daily lives. For example, a participant provided the following details about their journey with acceptance and self-forgiveness to cope with long COVID:

> "I found that my meditation practice has been incredibly helpful. Just acknowledging I'm not feeling well and it's okay if I'm not feeling well. And it's okay to stop and I don't have to catch up. It's okay. Just kind of allowing myself to kind of forgive myself for not being at the level that I'm normally at. So I find that really, really, very, very helpful to me." (Pre-MH)

Others emphasized the importance of gradually reincorporating physical exercises as a coping mechanism. Benefits associated with physical activity and time spent in nature were frequently mentioned, with one participant sharing: "For me, going out for walks and playing some sports to relieve the stress and anxiety." (No pre-MH) Meditation was another common practice among those with lived experience of long COVID, as is illustrated by the following contribution:

> "I do sleep meditations every night before I go to bed and that really helps, because I find I get all clenched up with my anxiety and so when I am lying in bed I just do things like clench your fingers, release your fingers, clench your arms, and then it goes through the body and it helps you relax." (Pre-MH)

Some participants also reported varying successes and challenges with the adjustment of behaviours related to nutrition, substance use, and distraction as coping mechanisms. For instance, many participants noted benefits associated with healthier diets and substance use elimination trials. In contrast, others reported increased indulgence in alcohol, cannabis, food, and sources of distraction. This latter perspective is represented in the following contribution:

> "I wanted to go back to smoking. If I would've had enough energy to get up I probably would've drank. So instead I ate and gained 30 pounds which is really shitty, but I went to the food and now trying to get that off. . .that actually besides the anger probably impacted my mental health the most, is those 30 pounds. So it's a bit of a different addiction but it's still an addiction." (Pre-MH)

**Social support and long COVID knowledge sharing.** Many participants reported the value of support and knowledge sharing among people living with long COVID as a coping mechanism. For instance, one participant shared:

> "I literally probably wouldn't be alive if it wasn't for the support groups and the sharing of information and support. And again, just feeling like I'm not alone in this. . .it's not a good thing because there's so many people in this position, but [I] have got a lot of strength and lifesaving information multiple times from those groups." (Pre-MH)

Though support from peers helped some participants to feel less alone and provided them with valuable information, others acknowledged the vast amount of information and experiences shared by peers on social media could be overwhelming. This experience is illustrated by the following contribution:

> "I'm on the long COVID support groups on Facebook but it's kind of all over the map. . . I find sometimes it's way too much information and it's not a regulated source of information. So you don't know what's good and what's bad with the information." (No pre-MH)

Some participants also mentioned support from friends, family, and community as beneficial to their coping experience, whether through socialization or assistance with daily living. One participant acknowledged social connections as central to their coping experience, stating: "The key is really to share with other people and to socialize and to not isolate one's self." (No pre-MH) Meanwhile, another participant noted the value of connection to plants and animals in addition to support from friends and family:

> "Even though I am anxious and I do feel angry all the time as I mentioned earlier, I do feel like I am able to cope okay–I think, based on everything else going around me. I have very supportive friends and very supportive family, and like I said, my husband's amazing. And I have all these animals and garden and things like that and the gratitude that I feel." (No pre-MH)

## Discussion

This study explored the experience of mental health, quality of life, and coping among people living with long COVID. The following five themes were recognized as central to the lived experience of long COVID: The Emotional Landscape of Long COVID, New Limits to Daily Functioning, Grief and Loss of Former Identity, Long COVID-related Stigmatization, and Learning to Cope with Persisting Symptoms. Though participants were initially assigned to separate focus groups based on whether they had pre-existing mental health conditions, major findings regarding the substantial impact of long COVID on mental health and wellbeing were consistent between groups. These themes illustrate the profound and enduring impact of long COVID on daily life, to the extent that many participants experienced a sense of shattered identity due to the vast emotional, physical, and social impact of their experiences. Despite these challenges, the extensive discussion of coping and help-seeking behavior among participants illustrates their commitment to reckoning with their new identity and advocating for an improved experience for all people living with long COVID.

In this study, participants reported the emergence of new or worsening mental health conditions as one of the most substantial changes associated with long COVID. This finding is consistent with a growing body of literature documenting the relationship between long COVID and mental health [12, 13, 30]. Though evidence indicates the negative influence of the COVID-19 pandemic on mental health for those with and without pre-existing mental health conditions alike [31, 32], little is known about mechanisms linking long COVID and mental health symptoms [33]. This is a notable gap in the literature that must be addressed to inform more effective supports and services.

Participants described the emotional harm caused by the stigmatization of long COVID, as well as its compounding effect on stigma against mental health conditions more generally. This is supported by the pre-existing literature, which has documented the association between higher rates of social stigma experienced by people living with long COVID and perceived anxiety, depression, stress, and reduced mental health-related quality of life [34]. According to participants, there are a range of stigmatizing beliefs about long COVID such as the notion that long COVID is not real, that individuals are to blame for their own illness, and that long COVID does not cause severe or persistent symptoms. Participants noted these ideas were first communicated by healthcare providers and the general public, but were eventually internalized. This stigma surrounding long COVID is akin to other poorly understood conditions, and may emerge from the disconnect between lived experience reports of severe symptoms and a lack of formalized knowledge regarding the origin, diagnosis, and course of the illness

[35, 36]. This indicates a need for improved understandings of the etiology of long COVID through research and clinical practice. In the meantime, efforts must be made to decrease the stigma surrounding long COVID in clinical settings and amongst the general public.

Our findings also emphasize the value of informal coping mechanisms. Though participants experienced some success with coping, many felt frustrated with the process of researching and implementing strategies, as a lack of credible information led to seemingly endless trials of ineffective and costly attempts to mitigate symptoms. While a recent systematic review of clinical trials highlights some emerging efforts to study interventions for mental health, cognition, and psychological wellbeing among people living with long COVID [37], this research remains limited, and inquiry into the effectiveness of various informal coping mechanisms for this population is warranted. In the absence of research on long COVID and coping, insight can be drawn from the chronic disease literature. Considering the persistent nature of long COVID, holistic approaches to coping with chronic disease may be an effective means of enhancing adaptive coping among people living with long COVID [38]. One example of this type of approach is the Optimal Health Program, a self-management program with evidence in the chronic disease sphere, which is currently being trialed in the UK for individuals with long COVID [39].

Another direction for future inquiry is to explore how the experience of mental health and quality of life changes over time for those living with long COVID. Since long COVID has only recently emerged, much of what is known about the condition has been derived retrospectively rather than prospectively [40]. Though our study included participants at various stages of their long COVID journey, a direct comparison of experiences based on this variable was not conducted. To better understand the progression of mental health and quality of life among people living with long COVID, a longitudinal design is necessary. Considering the frequent mention of long COVID as a disruptor to identity and source of uncertainty for the future among those with lived experience in this study, the work of Michael Bury [41] on chronic illness as a 'biographical disruptor' may provide a useful theoretical framework for these longitudinal investigations. Many participants also acknowledged the influence of socioeconomic status and access to relevant supports and services on long COVID recovery, indicating these may be important determining factors to investigate in future studies.

## Strengths and limitations

Several strengths and limitations must be acknowledged. This study was strengthened by the participation of the lived experience advisory group, as they contributed valuable perspectives and expertise to aspects of study design and execution. Another strength of this study is that efforts to maximize the diversity of the sample resulted in the inclusion of a variety of perspectives on the long COVID experience. Even so, it must be acknowledged that some voices may not be represented. The remote delivery of this study has significant advantages and disadvantages. Though it bolstered demographic diversity and maximized sample size by allowing individuals to participate from across Canada, it would have excluded people without access to internet or those with low technological literacy. A reliance on online recruitment may also have influenced participant responses to select questions, such as the use of online long COVID support groups as a coping mechanism. Lastly, approximately half the participants had not received an official diagnosis of long COVID at the time of enrollment. Given the limited testing for long COVID, self-reported symptoms were compared against the WHO long COVID diagnostic criteria [3] to assess eligibility, without excluding those who could not obtain an official diagnosis.

## Conclusion

Participant perspectives on the lived experience of long COVID indicate a life-changing impact of the condition on their mental health and quality of life. Though the type and severity of long COVID symptoms varied, feelings of loss of sense of self at the personal and interpersonal level permeated the participant experience. Declining emotional health and wellbeing compounded by reduced physical and cognitive capacities limited the ability of participants to engage with components of their former identity, including productivity, pursuits of joy, and the fostering of social connections. Amidst all these changes, the invalidation and stigmatization of the long COVID experience by others deepened feelings of isolation and hopelessness. While participants reported some success with informal coping mechanisms, many expressed fears of how their condition may deteriorate over time without formal supports and services to address symptoms. As the world begins to recover from the COVID-19 pandemic, we must promote the voices and prioritize the needs of those who continue to experience long-term impacts of the virus in research, clinical, and personal practice.

## Acknowledgments

We thank Jessie Tochez-Alvarez, Susan Deuville, Susie Goulding, Joel Semchuk, and additional members of the long COVID lived experience advisory group for their contributions to this study. We further thank Hamer Bilbao for supporting this project as focus group co-facilitator.

## Author Contributions

**Conceptualization:** Colleen E. Kennelly, Anh T. P. Nguyen, Gillian Strudwick, Chantal F. Ski, David R. Thompson, Mary Bartram, Sophie Soklaridis, Susan L. Rossell, David Castle, Lisa D. Hawke.

**Formal analysis:** Colleen E. Kennelly, Anh T. P. Nguyen, Natasha Yasmin Sheikhan, Lisa D. Hawke.

**Funding acquisition:** Natasha Yasmin Sheikhan, Gillian Strudwick, Chantal F. Ski, David R. Thompson, Mary Bartram, Sophie Soklaridis, Susan L. Rossell, David Castle, Lisa D. Hawke.

**Investigation:** Anh T. P. Nguyen, Natasha Yasmin Sheikhan, Lisa D. Hawke.

**Methodology:** Colleen E. Kennelly, Natasha Yasmin Sheikhan, Gillian Strudwick, Chantal F. Ski, David R. Thompson, Mary Bartram, Sophie Soklaridis, Susan L. Rossell, David Castle, Lisa D. Hawke.

**Project administration:** Anh T. P. Nguyen, Lisa D. Hawke.

**Resources:** Lisa D. Hawke.

**Supervision:** Lisa D. Hawke.

**Validation:** Lisa D. Hawke.

**Writing – original draft:** Colleen E. Kennelly.

**Writing – review & editing:** Anh T. P. Nguyen, Natasha Yasmin Sheikhan, Gillian Strudwick, Chantal F. Ski, David R. Thompson, Mary Bartram, Sophie Soklaridis, Susan L. Rossell, David Castle, Lisa D. Hawke.

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
