## [Decision Letter · Decision Letter 0]

19 May 2023

PONE-D-23-12139The lived experience of long COVID: A qualitative study of mental health, quality of life, and copingPLOS ONE

Dear Dr. Hawke,

Thank you for submitting your manuscript to PLOS ONE. After careful consideration, we feel that it has merit but does not fully meet PLOS ONE’s publication criteria as it currently stands. Therefore, we invite you to submit a revised version of the manuscript that addresses the points raised during the review process.

Please check for typos and consider the suggestions.   A very nice paper and we look forward to receiving your revisions. 

We look forward to receiving your revised manuscript.

Kind regards,

Julia Morgan

Academic Editor

PLOS ONE

Journal Requirements:

"David Castle has received grant monies for research from Servier, Boehringer Ingelheim; Travel Support and Honoraria for Talks and Consultancy from Servier, Seqirus, Lundbeck. He is a founder of the Optimal Health Program (OHP), and holds 50% of the IP for OHP; and is part owner of Clarity Healthcare. He does not knowingly have stocks or shares in any pharmaceutical company. Other authors have no conflict of interest to declare."

Additional Editor Comments:

Thank you for submitting this very nice paper which we all enjoyed reading. There are a few minor revisions that would improve the paper please correct any typos and consider the suggestions. I look forward to receiving your revisions.

Reviewers' comments:

Reviewer's Responses to Questions

**Comments to the Author**

1. Is the manuscript technically sound, and do the data support the conclusions?

Reviewer #1: Yes

Reviewer #2: Yes

2. Has the statistical analysis been performed appropriately and rigorously? 

Reviewer #1: N/A

Reviewer #2: N/A

3. Have the authors made all data underlying the findings in their manuscript fully available?

Reviewer #1: Yes

Reviewer #2: Yes

4. Is the manuscript presented in an intelligible fashion and written in standard English?

Reviewer #1: Yes

Reviewer #2: Yes

5. Review Comments to the Author

Reviewer #1: Please see the attached file. I enjoyted reading your well-written paper. My comments are brief, buit I wanted to be able to use bold and italics to demonstrate a suggestion I have to you, and could not do that through the online form.

Reviewer #2: This is a very well written paper of a well executed study which will make a valuable contribution to the literature. It needs a good read through for small grammar and tense issues in places --

For example p 3 end of last para—of cough not just cough

Tense p9 above research team positionality heading--- included not include

plural Participants bottom page 12

6. PLOS authors have the option to publish the peer review history of their article (what does this mean?). If published, this will include your full peer review and any attached files.

Reviewer #1: No

Reviewer #2: No

---

## [Author Response · Author response to Decision Letter 0]

23 May 2023

Reviewer #1: Please see the attached file. I enjoyted reading your well-written paper. My comments are brief, buit I wanted to be able to use bold and italics to demonstrate a suggestion I have to you, and could not do that through the online form.

We thank the reviewer. Please see below.

Reviewer #2: This is a very well written paper of a well executed study which will make a valuable contribution to the literature. It needs a good read through for small grammar and tense issues in places --

For example p 3 end of last para—of cough not just cough

Tense p9 above research team positionality heading--- included not include

plural Participants bottom page 12

We thank the reviewer for these comments. We have proofed thoroughly and corrected the items identified.

The lived experience of long COVID

Comments to the authors

Thank you for the opportunity to review this manuscript, reporting a qualitative study of the lived experience of long COVID. It really is a beautifully written piece, and I have no requests for any major amendments. The study design, including recruitment, ethics, data collection and analysis is well described, and it is refreshing to see open acknowledgement of the contribution of the advisory group to the study. The findings are impactful and shine a spotlight on the plight of Canadians living and struggling with the consequences of long COVID. 

We thank the reviewer for the review of our manuscript and the positive comments. We have responded to the comments below and in text.

The only tiny little suggestions I would make would be:

1. To present the theme labels in italics in the text wherever these are first mentioned and capitalise the first word of each theme, and 

We have made this revision throughout.

2. to avoid the word ‘include ‘ since it suggests there are other sub-themes that are not being mentioned. 

We have clarified each such statement.

For example, on p12: Theme 1: The emotional Landscape of long COVID. At the end of this introductory paragraph, you write: ‘Subthemes which represent this experience include: anxiety and fear of the unknown, challenges with emotional regulation, and feelings of hopelessness and depression’. That is quite tricky for the reader to unpick, but if you did this (and applied it throughout wherever appropriate), you would make things much easier for the reader: 

‘Three subthemes represent this experience: Anxiety and fear of the unknown, Challenges with emotional regulation, and Feelings of hopelessness and depression’ … 

Thank you again – it has been a pleasure to review this work. 

We thank the reviewer for illustrating their request and have made these changes.

---

## [Editor Report · Decision Letter 1]

25 Sep 2023

The lived experience of long COVID: A qualitative study of mental health, quality of life, and coping

PONE-D-23-12139R1

Dear Lisa Hawke,

We’re pleased to inform you that your manuscript has been judged scientifically suitable for publication and will be formally accepted for publication once it meets all outstanding technical requirements.

Kind regards,

Julia Morgan

Academic Editor

PLOS ONE
---

## [Editor Report · Acceptance letter]

6 Oct 2023

PONE-D-23-12139R1 

The lived experience of long COVID: A qualitative study of mental health, quality of life, and coping 

Dear Dr. Hawke:

I'm pleased to inform you that your manuscript has been deemed suitable for publication in PLOS ONE. Congratulations! Your manuscript is now with our production department. 

Kind regards, 

on behalf of

Dr. Julia Morgan 

Academic Editor

PLOS ONE